# Therapeutic Efficacies of Berberine against Neurological Disorders: An Update of Pharmacological Effects and Mechanisms

**DOI:** 10.3390/cells11050796

**Published:** 2022-02-24

**Authors:** Jia-Wen Shou, Pang-Chui Shaw

**Affiliations:** 1School of Life Sciences, The Chinese University of Hong Kong, Hong Kong 852852, China; shoujiawen@163.com; 2Li Dak Sum Yip Yio Chin R&D Centre for Chinese Medicine, The Chinese University of Hong Kong, Hong Kong 852852, China; 3State Key Laboratory of Research on Bioactivities and Clinical Applications of Medicinal Plants and Institute of Chinese Medicine, The Chinese University of Hong Kong, Hong Kong 852852, China

**Keywords:** berberine, neurological disorders

## Abstract

Neurological disorders are ranked as the leading cause of disability and the second leading cause of death worldwide, underscoring an urgent necessity to develop novel pharmacotherapies. Berberine (BBR) is a well-known phytochemical isolated from a number of medicinal herbs. BBR has attracted much interest for its broad range of pharmacological actions in treating and/or managing neurological disorders. The discoveries in basic and clinical studies of the effects of BBR on neurological disorders in the last decade have provided novel evidence to support the potential therapeutical efficacies of BBR in treating neurological diseases. In this review, we summarized the pharmacological properties and therapeutic applications of BBR against neurological disorders in the last decade. We also emphasized the major pathways modulated by BBR, which provides firm evidence for BBR as a promising drug candidate for neurological disorders.

## 1. Introduction

Neurological disorders refer to any dysfunctions of the nervous system, and mainly include Alzheimer’s disease (AD), Parkinson’s disease (PD), Huntington’s disease (HD), dementia, schizophrenia, anxiety, depression, epilepsy, traumatic brain injury (TBI), and brain tumor [1,2]. The burden of deaths and disabilities caused by neurological disorders has been increasing dramatically, ranking it as the leading cause of disability and the second leading cause of death worldwide [2,3]. The World Health Organization predicts that by 2040, as many developed countries’ populations age, neurological disorders will overtake cancer to become the second leading cause of death worldwide [4]. Nevertheless, there is no treatment that can cure neurological disorders, and the current treatments mainly target the amelioration of symptoms [5,6]. Berberine, a natural alkaloid, is mainly isolated from *Coptis chinensis*, *Berberis vulgaris*, *Hydrastis canadensis*, and *Phellodendron amurense* [7]. For over a thousand years, these herbs have been used for treating diarrhea without any obvious side effects in patients [8]. With the advances of pharmacological research, BBR has been considered as a promising multitarget drug (MTD) for treating neurological disorders. In this review, we summarize the in vivo research on BBR for treatment against neurological disorders in the last decade and provide our comments about the omnipotent effects of BBR.

## 2. Pharmacokinetic Characteristics of BBR

After oral administration, BBR was absorbed by the gastrointestinal tracts of mice, rats, hamsters, rabbits, and beagle dogs [9,10,11,12,13]; however, the bioavailability was quite low (below 1%) [12,14]. Then, BBR was mainly distributed into the liver, followed by other organs including the intestine, kidneys, muscle, lungs, brain, heart, and pancreas [11,15]. Notably, BBR can cross the blood–brain barrier [11,16], although the brain concentration of BBR after oral administration was quite low (around 1 ng per g of brain tissue) [11]. Then, the absorbed BBR underwent phase I metabolism reactions, including demethylation, demethylenation, and reduction to produce metabolites M1-6 [10,11]; phase 1 metabolites were usually mediated by chrome P450 enzymes (CYPs) such as CYP2D6, CYP1A2, and CYP3A4 [17]. In addition, nitroreductase from gut microbiota was reported to reduce BBR into an intestine-absorbable form—dihydroberberine (dhBBR) [10]. Phase II metabolites were formed by glucuronidation, sulfation, and methylation of phase 1 metabolites via UDP-glucuronosyltransferases (UGTs), sulfotransferases (SULTs), and catechol-O-methyltransferase (COMT) catalyzation [18,19]. BBR and its metabolites were mainly excreted through feces, followed by urine, and bile [18]. The metabolic characteristics of BBR are summarized in Figure 1.

## 3. Efficacies and Mechanisms of BBR on Neurological Disorders

Neurological disorders are not contagious, but greatly affect quality of life, as these diseases not only lead to neural damage but also influence an individual’s movement, speech, memory, intelligence, and much more [20,21]. BBR has emerged as a promising medication for combating neurological disorders. Here, we address the efficacies and mechanisms of BBR, as follows.

### 3.1. BBR on Alzheimer’s Disease

Alzheimer’s disease (AD) most often develops in people over 65 years of age and is characterized by memory loss and handicapped daily functions [22]. To date, the exact cause of AD has not been fully discovered, but it is believed that AD results from multiple contributing factors. Thus, there is no direct and effective treatment for AD. There are two main strategies for treatment. Firstly, inhibiting the activity of cholinesterase (ChE), an enzyme to catalyze the breakdown of acetylcholine (ACh) and other choline esters that function as neurotransmitters, is one of the potential therapeutic strategies based on the cholinergic hypothesis [23,24]. Secondly, it is important to reduce amyloid beta (Aβ) and Tau protein plaques, which may lead to neurofibrillary tangle formation, oxidation, inflammation, and excitotoxicity [25,26].

Due to its multifaceted nature, BBR has been shown to address AD mainly in two aspects: anti-ChE and anti-Aβ/Tau pathways (Figure 2).

#### 3.1.1. Inhibitory Effect of BBR on ChE

The cholinergic hypothesis states that a deficit in central cholinergic neurotransmission resulting from a loss of ACh contributes to pathological development [27]. ChE, including acetylcholinesterase (AChE) and butyrylcholinesterase (BChE), is responsible for hydrolyzing ACh into choline and acetic acid [28]. ChE inhibitors are effective medication for AD as they enhance central cholinergic function by inhibiting ChE activities, thereby increasing the availability of ACh to stimulate memory and learning ability in the brain [27]. In the streptozotocin-induced sporadic AD model and the heavy-metals-induced AD-like disease model, BBR maintained the ACh level by inhibiting AChE activity [29,30]. BBR has a large hydrophobic surface and a cation; thus, hydrophobic residues in AChE interacted with BBR to form a binding pocket, which accounts for the interaction between AChE and BBR [31]. However, there is no publication discussing the effect of BBR on BChE in vivo.

#### 3.1.2. Anti-Aβ and Tau Effects of BBR

The Aβ peptide, consisting of 39–43 amino acids, is derived from the abnormal processing of the amyloid precursor protein (APP), and the accumulation of Aβ peptide has been considered as a hallmark of AD pathogenetic development [32]. The enzymes α-secretase, β-secretase (also called BACE), and γ-secretase take active roles in the processing of APP [33]. Tau proteins within the brain cells of AD brains are misfolded and abnormally shaped, deposits of which form tangles within the neural cells [34]. In AD, it is common to find tau hyperphosphorylation and aggregation, thus losing its ability to maintain the microtubule tracks; as a result, tau dysfunction could lead to the retraction of neuronal processes and thus cell death [34].

The oral administration of BBR significantly ameliorated learning deficits and spatial memory retention in transgenic mouse models of AD (TgCRND8 mice, APP/PS1 mice, and 3×Tg AD mice) [35,36,37,38]. A mechanistic study showed that BBR significantly decreased the levels of C-terminal fragments of APP and the hyperphosphorylation of APP via the protein kinase B/glycogen synthase kinase 3 (AKT/GSK3) signaling pathway [35]. BBR also inhibited the activity of β/γ-secretases or suppressed PRKR-like endoplasmic reticulum kinase/eukaryotic translation initiation factor-2 α (PERK/EIF2α) signaling-mediated BACE1 translation to downregulate the Aβ level in the AD mouse hippocampus [36,37,39]. In addition, promoting the clearance of Aβ is another mechanistic aspect of BBR. To promote Aβ clearance, BBR activated the autophagic process through initiating the phosphoinositide 3-kinase (PI3K)/Beclin-1 pathway [38] or by inhibiting the mammalian target of rapamycin/P70 S6 kinase (mTOR/p70S6K) signaling [40]. Additionally, Aβ is toxic to neural cells, as it can cause pore formation resulting in ion leakage, disturb cellular calcium balance, and destroy membrane potential, thus leading to apoptosis, synaptic loss, and cytoskeleton disruption [41]. BBR is effective in preventing Aβ-induced damage to neural cells. Bilaterally injecting rats with Aβ induced learning and memory impairments, while BBR administration ameliorated Aβ-induced toxicity [42]. BBR showed this beneficial effect via modulating the Ca^2+^-activated K^+^ channel to maintain the optimal level of Ca^2+^ entry [42]. Moreover, BBR reduced Aβ-related oxidative and inflammatory damage. The antioxidant effect of BBR was exerted via downregulating reactive oxygen species (ROS) level, promoting the activity of glutathione (GSH), and inhibiting lipid peroxidation [43]. BBR also normalized the production of cytokines such as tumor necrosis factor α (TNFα), interleukin 12 (IL-12), IL-6, and IL-1β to retard inflammation [30]. In addition, exposure to Aβ could potentially lead to microglial activation, thereby triggering a detrimental neural response [44]. No in vivo function of BBR regarding microglial activation has been revealed; only an in vitro study indicated that BBR could inhibit Aβ-induced microglial activation via a silencing of cytokine signaling factor 1 (SOCS1)-dependent modulation of the microglial M1/M2 activated state [45].

Targeting Tau, BBR can reduce its hyperphosphorylation and increase its degradation. In 3×Tg AD mice, BBR improved the spatial learning capacity, memory retention, and the mechanism involved in reducing tau hyperphosphorylation via modulation of the AKT/glycogen synthase kinase 3β (GSK3β) pathway, enhancing autophagic flux, and increasing tau clearance through the PI3K/Beclin-1/B-cell lymphoma 2 (Bcl-2) pathway [46]. In APP/PS1 mice, BBR was found to suppress nuclear factor kappa-light-chain enhancer of the activated B cells (NF-κB) signaling pathway to limit tau hyperphosphorylation [43].

In conclusion, BBR exhibits therapeutic efficacy on PD mainly through the inhibition of ChE activity and suppression of Aβ- and Tau-induced toxicity. The downregulation of ChE activity by BBR contributes to increased ACh availability in the brain [29,30]. Both Aβ and Tau are toxic to neural cells via triggering oxidant, inflammatory, and even death signals, while BBR can degrade Aβ and Tau to ameliorate their toxicity [36,37,38,39,40,42,43,46] (Figure 2).

### 3.2. BBR on Parkinson’s Disease

Parkinson’s disease (PD) is a progressive neurodegenerative disorder characterized by the degeneration of dopamine (DA) and non-DA neurons, which could lead to tremors, rigidity, bradykinesia, and gait disturbance [47]. No cure has been discovered for treating PD and the current therapy mainly focuses on lessening neuron loss [8]. BBR showed beneficial effects against the chemical-induced PD model (Figure 2).

BBR protected neurons from apoptosis induced by 1-methyl-4-phenyl-1,2,3,6-tetrahydropyridine/probenecid (MPTP/P) through the downregulation of the Bcl2/Bcl-2-associated X protein (BAX) ratio [48], through AMP-activated protein kinase (AMPK)-dependent enhancement of autophagy [49,50], or by preventing NLRP3 inflammasome activation [50]. In the 6-hydroxydopamine-induced PD model, BBR reduced ROS production, caspase-3 activation, and subsequent neuronal death [51,52] BBR also increased the expression of tyrosine hydroxylase (TH), a rate-limiting enzyme for dopamine synthesis, to promote neurogenesis [48,50]. Additionally, a recent study demonstrated that BBR could ameliorate PD by regulating gut microbiota. BBR enhanced TH to produce L-dopa by triggering the biosynthesis of tetrahydrobiopterin in the gut microbiota and subsequently led to an increased brain dopa level, therefore improving brain function in MPTP-induced PD mice [53].

In addition, rotenone is also widely used to establish PD models [54], whereas the effect of BBR on the rotenone model is less well understood and controversial. There is no in vivo study of BBR on the rotenone-induced PD model. For the in vitro efficacy, Kysenius and colleagues claimed that the subtoxic nanomolar concentration (30 nM) of BBR could sensitize neurons to rotenone injury [55], while Han and colleagues found that BBR protected SH-SY5Y cells from rotenone injury by activating the antioxidant and PI3K/AKT signaling pathway [56].

Collectively, BBR maintains neural viability in PD models. BBR not only lessens neuron loss [48,49,50,51,52,53], but also promotes neurogenesis [48,50,51] (Figure 2).

### 3.3. BBR on Stroke

Stroke, also defined as a cerebrovascular accident, is one of the major causes of mortality and long-term disability, and it is induced by either inadequate focal blood flow or hemorrhage into the brain tissue or the surrounding subarachnoid space [57]. The current prevention or treatment of stroke includes primary prevention, recanalization and thrombolysis, neuroprotection, secondary prevention, and neurorepair [58,59]. Both pretreatment and post-treatment of BBR have shown prominent efficacies for stroke (Figure 3).

BBR was found to be a thrombin inhibitor and had the ability to inhibit thrombin-induced platelet aggregation in washed platelet samples in vitro [60]; however, there is no research exploring the thrombolysis effect of BBR in vivo.

Middle cerebral artery occlusion (MCAO) surgery has been widely used to establish a successful murine stroke model [61]. After cerebral infarction occurs, oxidative factors and proinflammatory cytokines are released, leading to ischemic neuronal death including apoptosis and necrosis [62]. Following ischemia and reperfusion, a cascade of inflammatory responses is triggered. The high-mobility group box 1 (HMGB1) protein is released from necrotic and dying neural cells, subsequently activating the NF-κB pathway, which is commonly used as an indicator of inflammation in stroke studies [63,64]. Then, TNFα, IL-1β, and IL-6 are activated [65,66]. Seven-day pretreatment of BBR prevented the translocation of NF-κB into the nucleus and the transcription of proinflammatory cytokines; consequently, the expression of proinflammatory factors such as TNFα, IL-1β, and IL-6 was downregulated and the expression of anti-inflammatory cytokines, including IL-10, was upregulated [67]. Inflammation in stroke could lead to the production of ROS [68]. Excessive ROS may cause severe damage to neural cells, and then cell death by either necrosis or apoptosis may be initiated [63]. BBR pretreatment lowered the increased level of MDA and enhanced the activities of antioxidases such as superoxide dismutase (SOD), catalase (CAT), peroxiredoxin, and NAD(P)H dehydrogenase quinone 1 (NQO1) [67,69]; in addition, the preadministration of BBR lessened neural cell apoptosis via decreasing caspase cascades (caspase-3 and caspase-9) and increasing Bcl-2 expression [70,71,72], and promoting the cell-survival-related pathways such as the phosphor activation of AKT and increase of ERK1/2 [70,73]. Moreover, BBR bound to the poly (A) tail on retinoblastoma mRNA to antagonize the mRNA degradation and upregulation of the retinoblastoma protein during ischemia/reperfusion, which in turn inhibited apoptosis and facilitated cell survival in the injured brain [74].

Additionally, post-treatment of BBR results in effects similar to pretreatment. BBR administered after MCAO surgery reduced the infarction volume in mice and rats [75,76,77]. BBR functioned as a potent anti-inflammatory agent for ameliorating focal cerebral ischemia injury by enhancing the IL-10 level [75] and downregulating NF-κB nuclear transposition [76]; BBR was also able to activate the upregulation of claudin-5 expression to reduce access to the blood–brain barrier [76]. Scavenging ROS also contributed to the effect of post-treatment with BBR. Our previous study found that BBR acted as a potent agonist of peroxisome proliferator-activated receptor delta (PPARδ) to increase nuclear factor (erythroid-derived 2)-like 1/2 (NRF1/2) and NQO1 to lower the ROS content in MCAO mice brains, thus exhibiting the neuroprotective effect of BBR [77]. Moreover, BBR is also beneficial for facilitating angiogenesis by modulating AMP-activated protein kinase (AMPK)-dependent M2 macrophage/microglial polarization, which promoted functionary recovery against ischemic stroke [78].

Stroke occurrence results in a detrimental impact on the brain. The most prominent efficacy of BBR on stroke is to reduce brain infarct damage, which is achieved by promoting thrombolysis [60], decreasing oxidative and inflammatory damage [67,69,75,76,77], reducing neural cell death [67,70,71,72,73,74], and facilitating angiogenesis [78] (Figure 3).

### 3.4. BBR on Huntington’s Disease

Huntington’s disease (HD), also known as Huntington’s chorea, is mostly inherited and mainly characterized by chorea, dystonia, loss of motor coordination, and mental deterioration [79]. HD results from an expanded CAG repeat in the huntingtin gene, which encodes an abnormally long polyglutamine repeat in the huntingtin protein [80]. BBR effectively improved motor function and prolonged the survival rate of transgenic N171-82Q HD mice by increasing autophagic function to reduce mutant huntingtin accumulation [81].

### 3.5. BBR on Dementia

Dementia describes a group of symptoms regarding memory loss and thinking disability. It is not a specific disease, but brain disorders and aging have been confirmed to give rise to dementia [82,83]. The most common cause of dementia is AD, which accounts for 60–70% of dementia cases worldwide [82]. Vascular dementia accounts for at least 20% of dementia cases, making it the second most common type [84].

BBR treated AD-related dementia mainly via targeting AD symptoms (shown in Section 3.1). Moreover, BBR is also effective in treating vascular dementia, which is usually caused by reduced blood flow to the brain [84]. In the chronic cerebral hypoperfusion (CCH)-induced vascular dementia model, BBR treatment prevented cognitive deficits and reversed CCH-induced neuronal cell death [85]. In the diabetes-related vascular dementia model, BBR increased the blood supply from the posterior cerebral artery, which was achieved by the inhibition of miR-133a ectopic expression in the vascular endothelium and by the normalization of vascular bioactivity in the cerebral middle artery [86]. Neurotoxic chemicals, such as doxorubicin, d-galactose, and lipopolysaccharide also led to cognitive impairment. BBR significantly improved cognitive disability in doxorubicin- or lipopolysaccharide-treated rats, and also improved the mechanism of the antioxidant and anti-inflammatory effect [87,88]. BBR diminished oxidative stress through enhancing glutathione peroxidase (GPx), SOD, CAT, and GSH; additionally, BBR attenuated inflammation, as evidenced by the downregulation of cyclooxygenase 2 (COX-2), NF-κB, TLR4, TNFα, and IL-6 levels [87,88]. In d-galactose-induced dementia rats, BBR ameliorated memory loss by restoring the Arc expression level, which is a pivotal mediator in maintaining normal synaptic plasticity [89].

In conclusion, BBR restores normal brain functions in dementia subjects by increasing the brain blood supply [86], reducing oxidative and inflammatory damage [87,88], and maintaining normal synaptic plasticity [89].

### 3.6. BBR on Psychiatric Disorders and Epilepsy

Psychiatric disorders are mental illnesses that greatly disturb thinking, moods, and behaviors, which may increase the risk of disability, pain, and even death [90,91]. Major psychiatric disorders include schizophrenia, anxiety, and depression [92]. Moreover, these disorders are considered as comorbidities in epilepsy patients, as clinical evidence has shown a much higher rate of psychiatric disorders in epilepsy patients than in the healthy control group [93,94]. BBR produces prominent effects on psychiatric disorders and epilepsy (Figure 4).

#### 3.6.1. Schizophrenia, Anxiety, and Depression

Schizophrenia manifests as continuous or relapsing episodes of psychosis, and major symptoms include altered perceptions, abnormal thinking, and odd behaviors [95]. MK-801 (an NMDA receptor antagonist) administration results in schizophrenia-like behaviors in rodents; BBR treatment improved learning impairments, while the mechanism remains unexplored [96].

Anxiety and depression are interrelated and mutually influenced. Patients with depression often have anxiety disorders, and those with anxiety disorders commonly show depression features [97]. The causes of anxiety and depression are multiple, and factors such as chemical imbalance, environment, and heredity may play roles [98]. BBR is beneficial for anxiety and depression, as shown in Figure 4. Drug addiction, such as methamphetamine and morphine, can lead to anxiety and depression, while BBR remarkably attenuates this discomfort [99,100,101]. In morphine-addicted animals, BBR modulated the central noradrenergic system through restoring the decreased brain-derived neurotrophic factor (BDNF) level in the hippocampus and by suppressing locus coeruleus activation [99]. In ameliorating methamphetamine-induced anxiety, BBR not only lessened neuroinflammation by reducing TLR4 and NF-κB activation [100], but also increased oxytocin receptors in the nucleus accumbens and in the hippocampus to a lower oxytocin level [101], which is closely related to drug abuse [102]. In addition, clinical evidence shows that menopausal transition leads to an elevated risk of anxiety and depression [103,104,105]. BBR produced antidepressant-like effects in ovariectomized mice, which was achieved via the BDNF/cAMP-response element binding protein (CREB)/eukaryotic elongation factor 2 (eEF2) pathway-dependent activation of the 5-hydroxytryptamine 2 (5-HT_2_) receptor [106]. Moreover, BBR is also capable of modulating the gut microbiota to treat anxiety; in ovariectomized rats, BBR promoted the abundance of beneficial gut microbes, such as Bacteroides, Bifidobacterium, Lactobacillus, and Akkermansia, and increased equol generation to treat postmenopausal symptoms of anxiety [107].

#### 3.6.2. Epilepsy

Epilepsy is a common chronic neurological disorder, the hallmark of which is recurrent and unprovoked seizures [108]. Mutations in syntaxin 1b (Stx1b), encoding a presynaptic protein, cause fever-associated epilepsy syndromes [109]. Pentylenetetrazole caused decreased Stx1b, which induced seizure, whereas BBR showed the ability to increase the Stx1b level to inhibit seizure development [110]. In addition, excitotoxicity, neuroinflammation, and oxidative stress characterize the epileptogenic process, and these three aspects are considered as treatment targets [111]. The anti-inflammatory and antioxidant effect has been well documented in treating epilepsy. The ability of BBR in combating oxidative damage in epilepsy was exerted through reducing ROS, lipid peroxidation, and MDA levels, and by promoting the expression of antioxidases such as NRF2, CAT, SOD, and GPx [112,113,114,115]. The anti-inflammatory actions of BBR were related to a significant reduction in the recruitment of macrophages and neutrophils as well as levels of TNFα, IL-1β, and IL-6 [112,116].

Conclusively, BBR treats anxiety, depression, and epilepsy mainly through diminishing oxidative and inflammatory damage [100,112,113,114,115,116]. Additionally, the regulation of hormones (oxytocin), neurotransmitter-related targets (5-HT_2_, Stx1b), and gut microbiota also contribute to the efficacies of BBR [99,101,106,107].

### 3.7. BBR on Traumatic Brain Injury

Traumatic brain injury (TBI) is an injury to the brain caused by an external force, and it can result in bruising, torn tissues, bleeding, and other physical damage to the brain, which might subsequently cause long-term complications or death [117]. TBI leads to neurological disability due to primary and secondary injury mechanisms [118]. The primary injury occurs during the initial insult, while the secondary injury is due to the pathological changes that follow the insults [118]. The secondary injury affects the recovery outcome post-TBI, and the post-treatment of BBR has shown good efficacy in attenuating secondary injury. BBR reduced cortical lesion size and neuronal death by inhibiting microglia and astrocyte activation in both the cortical lesion border zone (LBZ) and the ipsilateral hippocampal CA1 region, and by inhibiting inducible nitric oxide synthase (iNOS) and COX-2 expression, thus suppressing the following oxidative and inflammatory injury [119]. In addition, the post-injury administration of BBR was found to be related to the inhibition of the TLR4/MyD88/NF-κB signaling pathway, which suppressed the inflammatory cascade in glial cells to ameliorate TBI [120]. Thus, the antioxidant and anti-inflammatory properties of BBR [119,120] contribute to its efficacies.

### 3.8. BBR on Tumor

Brain tumors occur due to a mass or growth of abnormal cells in the brain. Brain tumors can begin in the brain (primary brain tumors), or cancer in other body parts may spread to the brain as secondary (metastatic) brain tumors [121]. BBR can suppress various kinds of tumors, including brain tumors (Figure 5).

Gliomas account for nearly 70% of malignant primary brain tumors in adults, and the prognosis is quite poor [122]. BBR has emerged as a promising antiglioma medication via promoting cell death, senescence, and inhibiting angiogenesis and drug resistance.

BBR induced glioblastoma cell apoptosis through autophagy activation, which was achieved by the inhibition of the AMPK/mTOR/unc-51-like kinase 1 (ULK1) pathway [123]. In addition, BBR treatment could lead to glioblastoma cell oncosis [124], which is a noncanonical form of programmed cell death resulting from a rapid decrease in intracellular adenosine triphosphate (ATP) and mitochondrial dysfunction [125]. BBR reduced the oxygen consumption rate and inhibited mitochondrial aerobic respiration by repressing phosphorylated ERK1/2 (p-ERK1/2), thereby triggering oncosis-like cell death [124]. The induction of cellular senescence is another antiglioma mechanism of BBR, which is likely mediated by the downregulation of the epidermal growth factor receptor (EGFR)/ Raf-1 Proto-Oncogene (RAF)/mitogen-activated protein kinase (MEK)/ERK pathway [126].

Angiogenesis refers to the formation of new blood vessels, and it does not cause malignancy itself but can promote tumor progression and metastasis [127]. The antiangiogenesis effect of BBR was evidenced by the decreased level of hemoglobin and cluster of differentiation 31 (CD31) mRNA, proving that BBR reduced vascular density in glioma; this occurred by inhibiting the phosphorylation of vascular endothelial growth factor receptor-2 (VEGFR2) and ERK [128].

The efficacy of chemotherapy might be hampered by the development of therapeutic resistance in glioma [129]. BBR enhanced the sensitization of glioma against temozolomide (a chemotherapeutic agent) [130]. BBR efficiently increased glioma responses to temozolomide treatment, with a profound effect on the activation of the ERK1/2 pathway, triggering the autophagy and apoptosis processes [130].

In sum, the anti-brain-tumor action of BBR is mainly due to the inhibition of tumor growth, such as inducing cell death [123,124,126] and suppressing angiogenesis [128]. Additionally, BBR shows a synergic effect by enhancing chemotherapy efficacy [130].

## 4. Clinic Applications

Several clinical trials of BBR on stroke and schizophrenia patients have been performed and are summarized in Table 1.

### 4.1. Effect of BBR on Stroke Patients

BBR exerted a therapeutic effect against stroke mainly by ameliorating oxidative, apoptotic, and inflammatory damage. The common doses of BBR for treating stroke ranged from 0.9 to 2.1 g/day with a treatment period of two weeks [131,132,133,134]. BBR treatment could significantly improve neural function, reduce the plasma MDA level, and upregulate the GSH-Px level in plasma [131,132]. In addition, the levels of the macrophage migration inhibition factor (MIF), IL-6, HIF-1α, and caspase-3 were also reduced after BBR treatment, indicating BBR was beneficial for stroke recovery [133,134].

### 4.2. Effect of BBR on Schizophrenia Patients

There is no direct research on the clinical efficacies of BBR alone on schizophrenia. BBR has shown good efficacies when used in combination with risperidone, which is a first-line treatment for schizophrenia in a clinical setting. BBR not only enhances the action of risperidone but also ameliorates the side effects induced by risperidone. BBR (0.9 g/day) combined with risperidone (6 mg/day) were administered to patients. In comparison with patients treated with risperidone only, BBR significantly improved the learning memory function and the speed of information processing [135]. Then, BBR treatment corrected the endocrine hormone level disorder by decreasing the serum triiodothyronine level and by increasing the prolactin content; additionally, BBR attenuated oxidative stress caused by risperidone through increasing SOD, GSH-Px, and CAT expression and by decreasing the MDA level [136]. Another similar trial indicated that BBR was able to suppress inflammatory markers induced by risperidone, as evidenced by the downregulation of IL-1β, IL-6, and TNFα levels [137].

As metabolic syndrome is closely related to schizophrenia [140], the treatment of metabolic disturbances in schizophrenia has been well studied. An eight-week treatment of BBR (0.9 g/day) effectively decreased weight gain [138], levels of total cholesterol, low-density lipoprotein cholesterol, fasting serum insulin, and insulin resistance [139].

## 5. Concluding Remarks and Future Perspectives

In the last decade, plenty of studies have confirmed that BBR is beneficial for treating brain disorders. The omnifarious efficacy of BBR is mediated by its multitargeted mechanisms. From the above, we conclude that there might be four aspects of BBR to elucidate the multitargeted pharmacological effects (Figure 6).

### 5.1. Cell-Viability-Related Pathway

Modulating cell viability is the most obvious representation of the efficacy of BBR. BBR exerted influence on the cell-viability-related pathway by affecting cell proliferation, apoptosis, autophagy, as well as angiogenesis. BBR could maintain cell viability in AD, PD, stroke, dementia, and TBI, while in brain cancer, BBR treatment could provoke cell death. One chemical can trigger opposite effects according to different cell types and physiological status, which is considered as bidirectional regulation [141]. Many natural products share the bidirectional regulation effect, such as jaceosidin [142], curcumin [143,144], ginsenosides [145,146], and baicalin [147,148].

The efficacies of one chemical towards normal and tumor cells might be based on different factors. The major difference between normal and tumor cells is that tumor cells are immortal; consequently, tumor cells could suppress apoptosis by inhibiting tumor suppressor gene expression including p53, retinoblastoma protein, Bcl2, and TNF-related apoptosis-inducing ligand receptors [149,150]. Thus, for combating tumors, the activation of tumor suppressor genes could trigger apoptotic pathways to induce cell death, while normal cells showed less sensitivity to apoptotic signals [151,152]. Moreover, for diseases or chemical-induced toxicity to normal cells, the antiapoptotic effect of BBR was executed to protect them from damage. Overall, the anti-/proapoptotic effects could be interpreted as the protective effect of BBR for eliminating tumor cells or toxicity from the host. However, more research is needed to elucidate how BBR could sense different cell types and then trigger distinct pathways.

In brain disorders, neural cells are exposed to various stresses, including ROS, Aβ, Tau, neural toxic chemicals, and tumors, and are easily damaged. Regulating cell viability by BBR is manifested as affecting cell proliferation, apoptosis, autophagy, as well as angiogenesis. The mitogen-activated protein kinase (MAPK) signaling pathway is pivotal in regulating cell viability, including the ERK1/2, p38 MAPK, and JNK pathways [153]. It is well-documented that BBR exhibits interactions with ERK1/2. BBR modulated ERK1/2 phosphorylation to maintain cell viability in AD and stroke, and to provoke apoptotic death in brain tumors, showing the bidirectional regulation of BBR. The antiangiogenetic effect of BRB on brain tumors is also dependent on ERK1/2. Therefore, BBR has been shown as a potential autophagy modulator, as it could follow AMPK- or PI3K-dependent mechanisms to regulate autophagy, showing different efficacies against AD, PD, and brain tumors.

### 5.2. Oxidation- and Inflammation-Related Pathways

The coexistence of inflammation and oxidative stress responsible for neural pathological progressions has been well documented [154]. While inflammation and oxidative stress consist of distinct biochemical cascades, the processes are closely intertwined and function in parallel [154]. An unregulated imbalance in the host between the production of reactive chemicals and the elimination by antioxidases (refering to the protective effects) could lead to damaged important biomolecules and cells, which would have an impact on the whole organism and could cause many chronic diseases; during the damage, oxidative stress can instigate the generation of proinflammatory factors. Then, activated inflammatory cells would release mediators (cytokines, chemokines, nitric oxide, etc.) that induce tissue damage and, in turn, augment oxidative stress [155].

BBR is a promising antioxidant in various neurovegetative models, both in vitro and in vivo. The prominent action of BBR is to reduce ROS production. BBR scavenged ROS through upregulating antioxidases such as GSH, SOD, CAT, NQO1, NRF2, and GPx, which are enzymatic antioxidants and are capable of decomposing ROS [156]. In addition, overwhelmed ROS might trigger cell apoptosis, for instance, in stroke. Notably, BBR was also effective in counteracting ROS-induced apoptosis.

The regulation of the NF-κB pathway contributes to the anti-inflammatory effects of BBR. The inhibitory effect of BBR on NF-κB has been verified in stroke, dementia, anxiety, and TBI. NF-κB activation is tightly regulated, mainly through its localization. In resting cells, NF-κB proteins are kept in the cytoplasm, while activated forms of NF-κB mainly lie in the nucleus [157]. Then, activated NF-κB could execute its transcriptional function to mediate proinflammatory or anti-inflammatory gene expressions [157]. BBR is effective in preventing the translocation of NF-κB into the nucleus and the transcription of proinflammatory cytokines; therefore, proinflammatory genes were suppressed, and anti-inflammatory cytokines were promoted. Additionally, as oxidative stress and inflammation are intertwined, the antioxidant capability of BBR might contribute to its anti-inflammatory effect, and vice versa.

### 5.3. Gut-Microbiota-Related Pathway

The gut microbiota has been identified and proposed to be a key modulator of human health, to such an extent that it is considered as a hidden organ of the human body. Brain disorders are associated with gut microbiota dysbiosis, and the dysbiosis could also promote disease development [158,159]. This interplay is defined as the gut–brain axis [160]. It has been observed that significant changes in the microbial composition are apparent in the gut of AD, PD, and stroke patients [161,162,163]. The mediators between the brain and the gut microbiota mainly include short chain fatty acids (SCFAs), serotonin, gamma-aminobutyric acid (GABA), and inflammatory cytokines [164,165]. SCFAs could protect the blood–brain barrier, modulate the inflammatory cascade, affect the vagal nerve pathway, and activate the host immune system [166,167]. The gut microbiota regulates the level of serotonin In the colon and in the blood, and the alteration of serotonin-producing bacteria, showing the potential to treat serotonin-related diseases such as PD [165]. The gut microbiota also affects circulating GABA levels, which has been linked to cognitive impairment and AD development [168]. In addition, the inflammatory stage would be activated in the leaky gut, induced by microbiota dysbiosis via the release of inflammatory cytokines such as IL-6, IL-1β, and TNF-α [169]. Then, these cytokines would damage the brain barrier integrity and induce neuroinflammation via systemic routes [170]. So, would the gut microbiota play a role in BBR treatment of neurological disorders? This point will be further discussed in the next section.

### 5.4. Future Perspective

Overall, BBR seems to be a promising candidate for treating brain disorders. However, the safety of BBR should be emphasized. Clinical studies have revealed that repeated oral administration of BBR may inhibit the activities of CYP2D6, CYP2C9, and CYP3A4 in healthy volunteers [171], which may interfere with the metabolism of other drugs and would cause drug–drug interactions. In addition, there were some mild, transient, gastrointestinal adverse effects in T2DM patients after a 13-week BBR treatment, but no functional liver or kidney damage was observed [172]. As gut microbiota could affect the absorption of BBR [10], the coadministration of antibiotics (or antibiotic-like chemicals) with BBR should be used with caution.

Furthermore, little is known about why BBR could exert its various pharmacological activities. As shown in Figure 6, BBR could influence different pathways for treating brain disorders. More studies are needed to illustrate why BBR is omnipotent. For example, whether BBR could activate cell apoptosis or not depends on cell types, or whether there are any smart genes controlling this switch. Our recent research found that BBR functioned as a potent ligand to activate PPARδ to protect mice from stroke injury [77]. In addition, our previous research suggested that BBR was able to promote neurogenesis in vitro [173]. Considering that PPARδ, a ligand-inducible transcription factor, governs a variety of neural activities including cell differentiation, proliferation, and development [174], it is possible that PPARδ is a master gene in the BBR treatment of neurological disorders.

Owing to the extremely low bioavailability of BBR, another concern is whether there is a high enough amount of BBR in the brain to execute its pharmacological effects. A number of groups have found that BBR is prominent in attenuating cerebral ischemia injury [67,69,70,71,74,76,77,78]. In addition, our research suggested that the oral administration of BBR activated PPARδ to exert its protective action, and the binding affinity (Kd) between BBR and PPARδ is 290 nM [77]. However, the peak brain BBR concentration after oral administration ranges from 10 to 108 ng per g of brain tissue [11,175,176,177], and even via the intravenous route, the peak BBR concentration in the brain is only around 270 ng per gram of brain tissue [178]. Therefore, there may be endogenous factors/chemicals cooperating with BBR to exert its pharmacological effects. In treating PD, there are two aspects regarding the effects of BBR. First, BBR exhibits direct action on PD models, including the lessening of neuron loss and promoting neurogenesis [48,49,50,52,53]. Second, Wang and colleagues report that BBR ameliorated PD manifestation by upregulating the biosynthesis of L-dopa in the gut microbiota through a vitamin-like effect [51], suggesting an indirect effect of BBR. This kind of dual action is also found in hyperlipidemia. It is reported that BBR combated hyperlipidemia via the direct effect of the circulated BBR and the indirect effect of working through the butyrate of the gut microbiota [179].

To date, the role of BBR in treating PD and anxiety has been confirmed to relate to the gut microbiota [51,107]. It has been revealed that BBR not only modulates the gut microbiome structure, but also promotes some active microbial metabolites (e.g., L-dopa and equol). BBR has been well-documented in regulating SCFA and reducing gut leakage in metabolic syndrome [180]. Nevertheless, it is still unknown whether these effects could contribute to the efficacy of BBR in treating neurological diseases. More studies on the change in gut microbiota and metabolites upon the administration of BBR for neurological diseases is warranted.

## Figures and Tables

**Figure 1 cells-11-00796-f001:**
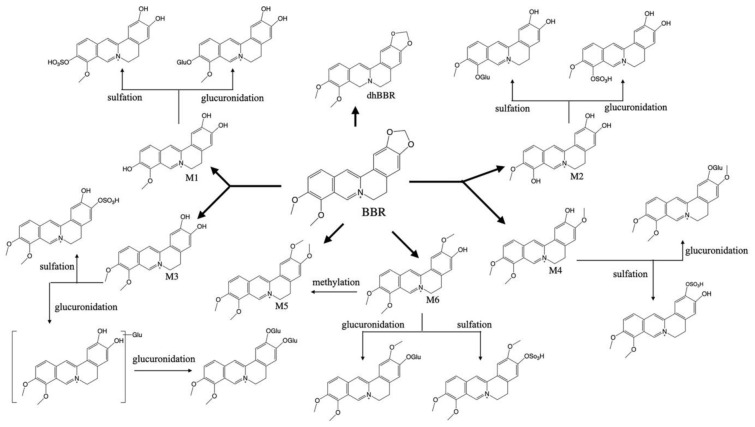
Metabolic pathway of BBR.

**Figure 2 cells-11-00796-f002:**
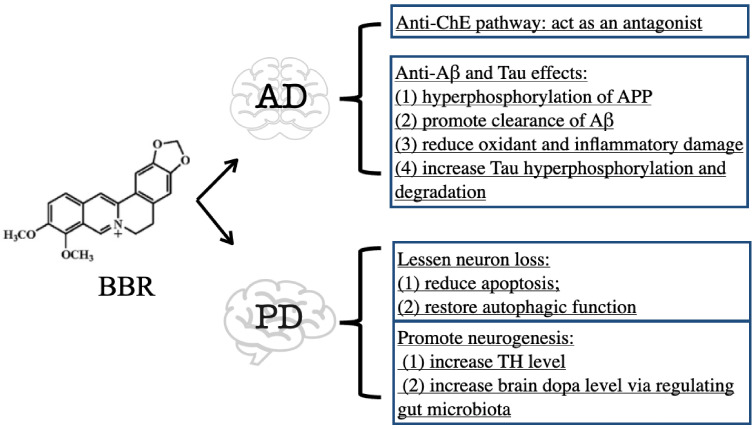
Effects of BBR against AD and PD.

**Figure 3 cells-11-00796-f003:**
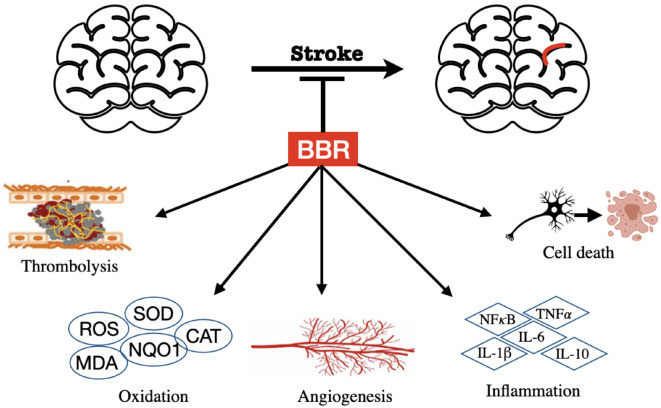
Effects of BBR against stroke.

**Figure 4 cells-11-00796-f004:**
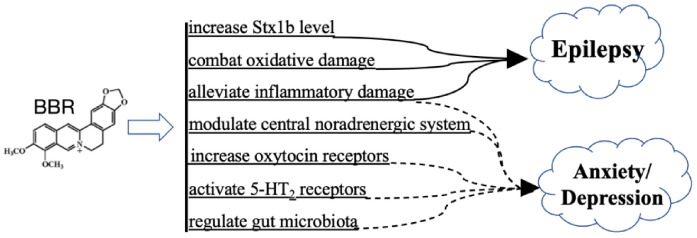
Effects of BBR against psychiatric disorders and epilepsy.

**Figure 5 cells-11-00796-f005:**
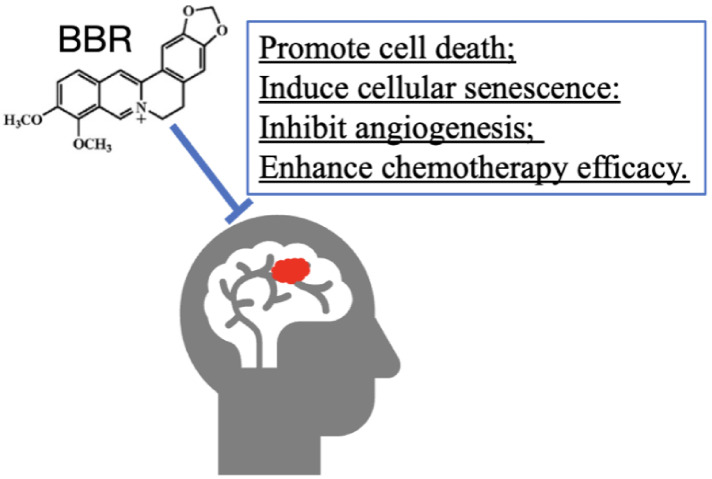
Effects of BBR against brain tumor.

**Figure 6 cells-11-00796-f006:**
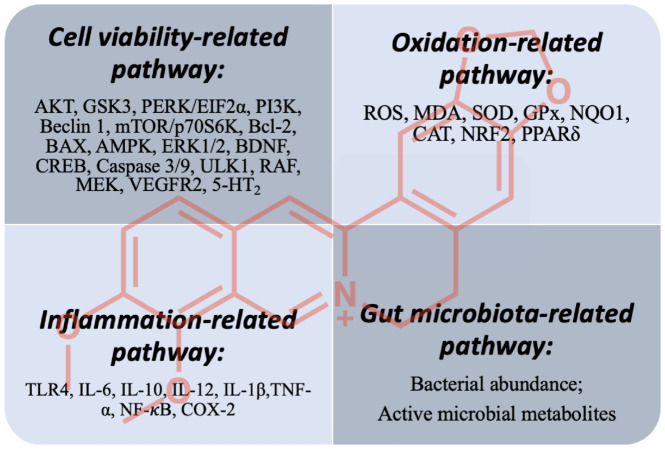
A summary of mechanisms of BBR on neurological disorders.

**Table 1 cells-11-00796-t001:** Clinical trials of BBR on stroke and schizophrenia patients.

Disease	No. of Patients	Dosage /Duration	Outcome	Ref.
Acute cerebral ischemic stroke	55	300 mg (tid) /14 days	Improved neural function; decreased MDA level; increased GSH-Px level	[131]
Acute cerebral ischemic stroke	52	500 mg (tid) /14 days	Improved neural function; decreased MDA level	[132]
Acute cerebral ischemic stroke	60	300 mg (tid) /14 days	Decreased levels of MIF and IL-6	[133]
Acute cerebral infarction	63	700 mg (tid) /7 days	Reduced serum HIF-1α, caspase-3 level, and fatality rate	[134]
Schizophrenia	31	300 mg (tid) /12 weeks	Improved learning memory function and information processing	[135]
Schizophrenia	43	300 mg (tid) /2 months	Increased prolactin, SOD, GSH-Px, and CAT levels; decreased MDA and triiodothyronine levels	[136]
Schizophrenia	34	300 mg (tid) /12 weeks	Decreased IL-1β, IL-6, and TNFα levels	[137]
Schizophrenia	27	300 mg (tid) /8 weeks	Decline in weight gain	[138]
Schizophrenia	27	300 mg (tid) /8 weeks	Decreased levels of total cholesterol, low-density lipoprotein cholesterol, fasting serum insulin, and insulin resistance	[139]

## Data Availability

Not applicable.

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
