# Peer review of "Therapeutic Efficacies of Berberine against Neurological Disorders: An Update of Pharmacological Effects and Mechanisms"

_cells, 2022, doi:10.3390/cells11050796_

Round 1
Reviewer 1 Report
The review by Jia-Wen Shou and Pang-Chui Shaw concerns the therapeutic effects of berberine in central neuropathologies. Honestly speaking I do not feel the review add new vistas or revise new important knowledge with respect to previous manuscript. Furthermore, some paragraphs (i.e. those concerning anxiety , epilepsy and brain injury) are very short and beside a brief description of the pathologies (largely insufficient by a general point of view) do not introduce specific knowledge concerning the therapeutic activities of berberine.
The title of paragraph is not consistent with content and must be changed. Furthermore the sub-paragraphs deal with some issues that are already introduced in previous chapters.
In general the impression is that there are no new entry concerning this natural compound to support the proposal of a new review
Author Response
Response: Thanks for your suggestion. We have merged several short paragraphs (Schizophrenia, anxiety and depression) and re-named them as psychiatric disorders.
We also revised the title of paragraph, for example, we replaced “3 Therapeutic actions of BBR against neurological disorders” with “3. Efficacies and mechanisms of BBR on neurological disorders”; and replaced “3.1 Alzheimer’s disease” with “3.1 BBR on Alzheimer’s disease”.
The most recent review of BBR concerned with neurological disorders was published in 2019 and only included Alzheimer’s, Parkinson’s and Huntington’s diseases [1]. To our best knowledge, it is our review that firstly and systematically discusses a broad range of neurological disorders including Alzheimer’s disease, Parkinson’s disease, Huntington’s disease (HD), dementia, schizophrenia, anxiety, depression, epilepsy, traumatic brain injury and brain tumor. We believe this review would boost our understanding of effects and mechanisms of BBR on these mentioned diseases. Besides, we also expanded the part of “5.5 Future perspective”. Considering the extremely low bioavailability, we have proposed that it is typical of BBR to show two modes of action: one is the direct effect of BBR, and the other is the indirect effect like regulating gut microbiome. Please refer to the revised manuscript (5.5) for details.
We have highlighted the revised parts in red and also performed language editing on MDPI website. Thank you.
Ref:
1 Fan, D., Liu, L., Wu, Z., & Cao, M. (2019). Combating neurodegenerative diseases with the plant alkaloid berberine: molecular mechanisms and therapeutic potential. Current neuropharmacology, 17(6), 563-579.
Reviewer 2 Report
The subject of the manuscript is interesting, nevertheless it raises some remarks
- What is the impact of BBR in rotenone models of Parkinson's disease?
- Considering that exposure to berberine increases the proportion of dopa/dopamine producing bacteria, what role could berberine play in Parkinson's disease via the gut microbiota?
- Since inflammatory process plays an important role in some neurological diseases, what is the effect of BBR on cyclooxygenase-2 expression?
- Do the authors have any insight regarding the effects of post-injury administration of berberine?
- Does berberine have an impact on the microglial activation state in the context of exposure to β Amyloid?
Author Response
The subject of the manuscript is interesting, nevertheless it raises some remarks
- What is the impact of BBR in rotenone models of Parkinson's disease?
Response: Rotenone is also widely used to establish PD model , whereas the effect of BBR on rotenone model is less understood and controversial. There is no in-vivo study of BBR on rotenone-induced PD model. As this review mainly focuses on in-vivo studies, we only briefly summarized the in-vitro efficacy of BBR on rotenone models (please refer to 3.4 for details in the revised manuscript). Kysenius and colleagues claimed that subtoxic nanomolar concentration (30 nM) of BBR could sensitize neurons to rotenone injury, while Han and colleagues found that BBR protected SH-SY5Y cells from rotenone injury through activating antioxidant and PI3K/AKT signalling pathway.
- Considering that exposure to berberine increases the proportion of dopa/dopamine producing bacteria, what role could berberine play in Parkinson's disease via the gut microbiota?
Response: L-dopa therapy has shown great efficacy in treating PD in clinic [1-2]. And a combination of carbidopa/L-dopa is widely used in clinic as it prevents L-dopa from being converted into dopamine prematurely in the bloodstream, allowing more of it to get to the brain [3]. According to Wang et al’s work [4], BBR exerted similar effects with L-dopa therapy against PD model but showed different mechanism. Wang et al found BBR enhanced tyrosine hydroxylase activity to produce L-dopa by triggering the biosynthesis of tetrahydrobiopterin in the gut microbiota. Thus, they concluded oral berberine improved brain dopa/dopamine levels to ameliorate Parkinson’s disease by regulating gut microbiota.
This point has been mentioned in 3.2 in our revised manuscript.
Ref:
1 Hornykiewicz, O. (2002). L-DOPA: from a biologically inactive amino acid to a successful therapeutic agent. Amino acids, 23(1), 65-70.
2 Cools, R. (2006). Dopaminergic modulation of cognitive function-implications for L-DOPA treatment in Parkinson's disease. Neuroscience & Biobehavioral Reviews, 30(1), 1-23.
3 Liao, X., Wu, N., Liu, D., Shuai, B., Li, S., & Li, K. (2020). Levodopa/carbidopa/entacapone for the treatment of early Parkinson’s disease: a meta-analysis. Neurological Sciences, 41(8), 2045-2054.
4 Wang, Y., Tong, Q., Ma, S. R., Zhao, Z. X., Pan, L. B., Cong, L., ... & Jiang, J. D. (2021). Oral berberine improves brain dopa/dopamine levels to ameliorate Parkinson’s disease by regulating gut microbiota. Signal transduction and targeted therapy, 6(1), 1-20.
- Since inflammatory process plays an important role in some neurological diseases, what is the effect of BBR on cyclooxygenase-2 expression?
Response: Thanks for your advice. We have omitted this important effect in our previous manuscript. And we have discussed this effect in the revised version. BBR is able to downregulate COX-2 expression in dementia (3.5) and traumatic brain injury (3.7).
- Do the authors have any insight regarding the effects of post-injury administration of berberine?
Response: Post-treatment of berberine is effective in treating stroke and traumatic brain injury. We have re-written the concerned paragraphs (3.3 and 3.7) to make it clear.
- Does berberine have an impact on the microglial activation state in the context of exposure to β Amyloid?
Response: Thanks for your advice. BBR is able to suppress microglial activation after exposure to β Amyloid and we have added this part in the revised manuscript (3.2).
We have highlighted the revised parts in red and also performed language editing on MDPI website. Thank you.
Reviewer 3 Report
In the proposed manuscript authors represent well informative review about treatment potential for various CNS disturbances applieing natural alkaloid berberine, and this summarised information could attract attention of respective scientific community.
Author Response
Thanks for the positive comments.
Round 2
Reviewer 1 Report
It is the opinion of this referee that the revision of the manuscript did not improved sufficiently its qaulity and interest. In the revised form tha largets part of teh text is still dedicated to describe the neurological diseases, in a very generic way honestly speaking, and few parts relate to the effects of berberine, possibly because the data so far available are insufficient. The interest of the review is tehrefore very limited and does not deserve to be published in teh submitted form
Author Response
Dear reviewer,
Thanks for your comment. To our best knowledge, it is our review that firstly and systematically discusses a broad range of neurological disorders including Alzheimer’s disease, Parkinson’s disease, Huntington’s disease (HD), dementia, schizophrenia, anxiety, depression, epilepsy, traumatic brain injury and brain tumor.
Though some parts contains several sentences regarding to some disease-related genes, for exmaple HMGB1 and Stx1b, it is necessary to explain these key genes, as they may play a key role in disease development or in the effect of BBR. Besides, in some diseases like Huntington's disease and Epilepsy, there are only a few studies, we believe that including these diseases is also of important and hope to serve as a modest spur to induce others to come forward with their valuable contributions.
Besides, we have added several sentences in each part to highlight the most important effects of BBR. Please refer to the revised manuscript for details.
Please re-consider our manuscript.
Thanks you.